# Deep Annotated Learning, Harmonic Descriptors and Automated Diabetic Retinopathy Detection

**J. Desbiens**
Intelligent Retinal Imaging System
jdesbiens@retinalscreenings.com

**S. Gupta, MD**
Retina Specialty Institute
sgupta@retinalscreenings.com

**J. Stevenson**
Intelligent Retinal Imaging System
jstevenson@retinalscreenings.com

**G.A. Alderman**
Intelligent Retinal Imaging System
aalderman@retinalscreenings.com

**A. Trivedi**
Microsoft
antriv@microsoft.com

**P. Buehler**
Microsoft
pabuehle@microsoft.com

## Abstract

In this article, we propose a multi-phase automatic grading system for Diabetic Retinopathy (DR). First, a pattern matching detector highlights the potential lesions on input colour fundus images. In a second phase, the annotated images are classified using a Convolutional Neural Network running over Microsoft Azure Machine Learning. The system achieved very high performance for referable DR on clinical datasets with accuracy up to 97.1%. Experiments conducted on eye blood vasculature reconstruction as a biomarker have showed a strong correlation between vasculature shape and later stages of DR.

## Introduction

Diabetic Retinopathy (DR) is the most common cause of blindness in the working population of the United States and Europe and it will become a more important problem worldwide. The World Health Organization (WHO) predicts that the number of patients with diabetes will increase to 366 millions in 2030. In patients with diabetes, early diagnosis and treatment have been shown to prevent visual loss and blindness [11].

Automated grading of DR has potential benefits such as increasing efficiency, reproducibility, and coverage of screening programs; reducing barriers to access; and improving patient outcomes by providing early detection and treatment. To maximize the clinical utility of automated grading, an algorithm to detect referable DR is needed.

Machine Learning has been leveraged for a variety of classification tasks including automated classification of DR. However, much of the work has focused on feature extraction engineering which involves computing explicit features specified by experts, resulting in algorithms built to detect specific lesions or predicting the presence of any type of DR severity. Deep Learning is a machine learning technique that avoids such engineering by learning the most predictive features directly from the images given a large data set of tagged examples.

1st Conference on Medical Imaging with Deep Learning (MIDL 2018), Amsterdam, The Netherlands.

Identifying candidate regions in medical images is of greatest importance since it provides intuitive illustrations for doctors and patients of how the diagnosis is infered. Recently, advances in Deep Learning have dramatically improved the performance of DR detection. Most of these Deep Learning systems treat Convolutional Neural Network (CNN) as a kind of black box, lacking comprehensive explanation. Our proposed system learns from image-level pre-processed annotations of DR highlighting the suspicious regions. It mimics the expert process of a clinician examining an image by selecting regions showing high probability of being lesions. Then annotated images are passed to a CNN which in turn predicts their respective DR severity.

We show that a combination of extracted features, targeting specific characteristics like micro-aneurysms, hemorrhages, and exudates for DR detection, with the robust potential of Deep Learning systems to characterize accurately all stages of DR without confusion from brightness or capture artifacts, yields more precise results. We coined the name "Deep Annotated Learning" (DAL) for this system.

## Clinical data

### Clinical dataset and image grading

The initial image population consists of a collection of fundus images coming from 250,000 adult diabetic patient exams (87,401 unique patients from various ethnic origins and U.S. locations) with 33,428 patients suffering from a sight threatening disease. Eye exams were done during normal routine checkups. Afterwards, all images were graded by ophthalmologists for the presence of diabetic retinopathy and diabetic macular edema. DR severity was graded according to the International Clinical Diabetic Retinopathy scale [1]. The severity breakdown per eye for the original population was: 62% Normal, 24% Mild, 8% Moderate, 3% Severe, and 3% Proliferative. To effectively create a valid dataset, images have been randomly chosen in the database based on their quality scores prior to being added to the process to create a stratified random sample. We employed an image quality process which provides a score for each image passed through it. This process was run on every processed image and a score is provided to show the expected gradeability of the image based on feature detection. Images of excellent, good, and adequate quality were considered gradable.

The graders for the training and validation sets were all US-licensed ophthalmologists.

## Development of the algorithm

### Radial symmetry transformation

Radial symmetry maps are generally used to detect interest points in images that correspond to object centers. Many works on radial symmetry maps exist but the most commonly used is the one of [13]. With this approach a symmetry score is calculated from votes of one pixel to surrounding pixels. The transform is calculated in one or more radii $n$, the value of the transform at radius $n$ indicates the contribution to radial symmetry of the gradients a distance $n$ away from each point

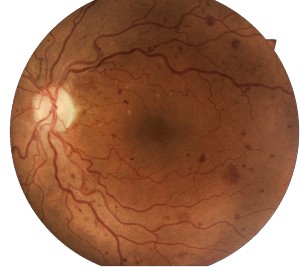 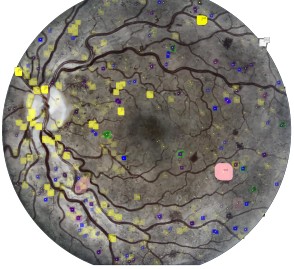

(a) An image of a proliferative DR retina taken by a CenterVue DRS retinograph

(b) Candidate bright and dark lesion ROI overlay on top of normalized image

Figure 1: Source and annotated images

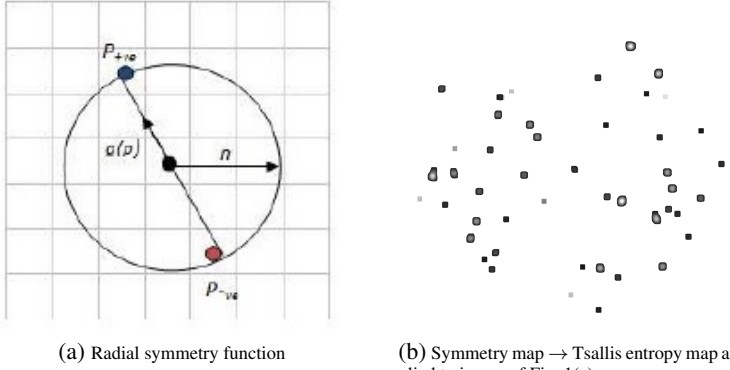

(a) Radial symmetry function

(b) Symmetry map → Tsallis entropy map applied to image of Fig. 1(a)

Figure 2: Image transformations

(see Fig. 2(a)). Round-shaped objects present in the image will appear as local optima in the result map.

**Entropy transformation**

In information theory, entropy is used to quantify the amount of information. The entropy reflects the information content of symbols independent of any particular probability model. Image analysis takes the concept of entropy in the sense of information theory, where entropy is used to quantify the minimum descriptive complexity of a random variable. Because the entropy can provide a good level of information to describe a given image, we compute the Tsallis' entropy of the distribution of the symmetry image and obtain an appropriate partition of the source image giving a probability map of the potential lesions (see Fig. 2(b)).

**Harmonic Descriptors**[1]

Consider a periodic signal $y(t)$ with period $T$. Since the period is $T$, we take the fundamental frequency to be $\omega_0 = \frac{2\pi}{T}$. We can represent almost all such function using the so-called *Fourier Series*.

$$y(t) = a_0 + \sum_{k=1}^{\infty} a_k \cos(k\omega_0 t) + b_k \sin(k\omega_0 t) = \sum_{k=-\infty}^{\infty} c_k e^{ik\omega_0 t} \text{ with } i = \sqrt{-1}.$$

The coefficients $a_k, b_k$ and $c_k$ being related by the following equations:

$$c_0 = a_0 \text{ and } c_k = \frac{a_k}{2} - i \cdot \frac{b_k}{2}, \text{ for } k \neq 0 \text{ (here } c_{-k} = c_k^*) \tag{1}$$

where it can be shown that:

$$a_0 = \frac{1}{T} \int_T y(t) dt, \text{ and } a_k = \frac{2}{T} \int_T y(t) \cos(k\omega_0 t) dt, b_k = \frac{2}{T} \int_T y(t) \sin(k\omega_0 t) dt \text{ for } k \neq 0,$$

and

$$c_k = \frac{1}{T} \int_T y(t) e^{-ik\omega_0 t} dt.$$

Usually, using complex Fourier expansion over real one is a matter of simplifying algebraic calculations.

---

[1]This section is based on Fourier Descriptors theory [21].

## Curve mapping

Let $\mathcal{C} = \{(x(t), y(t)) \mid t = 0, \ldots, N - 1\} \subset \mathbb{R}^2$ be a discrete curve that can be reduced to a discrete function $\phi : [0, N] \to \mathbb{R}$ after a rotation $\rho$ and a translation $\tau$. The coordinates can be then considered to be the sampling values

$$(x(t), y(t)) = (\tau^{-1} \circ \rho^{-1}) \left[ f\left(\frac{2\pi}{N}t\right) \right]$$

of a continuous curve $f : [0, 2\pi] \to \mathbb{R}$ that $f$ can be extended continuously to a $2\pi$-periodic function. When expanded into a Fourier series, a fixed number of discrete Fourier coefficients approximately represents the curve. Without loss of generality, we may consider from now on that we are dealing with periodic smooth periodic functions $\mathbb{R} \to \mathbb{R}$ having $[0, 2\pi]$ as their fundamental period $T$.

## Complex approximation

If $\mathbb{R}^2$ is seen as the complex plane $\mathbb{C}$, then the curve may be represented by a sequence of complex numbers $z(t) = x(t) + i \cdot y(t)$ having the discrete Fourier expansion $t = 0, \ldots, N - 1$:

$$z(t) = \sum_{k=0}^{N-1} \hat{c}_k e^{\frac{2\pi}{N}ikt} \text{ with } \hat{c}_k = \frac{1}{N} \sum_{t=0}^{N-1} z(t) e^{-\frac{2\pi}{N}ikt}.$$

When the coefficients $\hat{c}_k$ are interpreted as numerical approximations of the Fourier coefficients $c_k$ of the continuous curve $f : t \mapsto (\rho \circ \tau)\left(x(\frac{N}{2\pi}t)\right) + i \cdot (\rho \circ \tau)\left(y(\frac{N}{2\pi}t)\right)$ then

$$c_k = \frac{1}{2\pi} \int_0^{2\pi} f(t) e^{ikt} dt,$$

the relation between $\hat{c}_k$ and $c_k$ being given by

$$\hat{c}_k \approx c_k \text{ for } 0 \leq k < \frac{N}{2} \text{ and } \hat{c}_{N-k} \approx c_{-k} \text{ for } 1 \leq k < \frac{N}{2}. \tag{2}$$

In conclusion, we can interchangeably use the $\hat{c}_k$ coefficients for the $c_k$ coefficients, and the $c_k$ for the $\hat{c}_k$.

## Asymptotic behavior

By virtue of the Riemann-Lebesgue Lemma [16], $\lim_{|k| \to \infty} c_k = 0$, so terms for large values of $|k|$ are negligible and can be discarded. Removing higher frequencies in Eq. (2) is thus equivalent to omitting middle coefficients of $(c_0, c_1, \ldots, c_{N-1})$. Using Eq. 1, it can be easily shown that $a_0 = c_0$ and $a_k = c_k + c_{-k}$, $b_k = i \cdot (c_k - c_{-k})$ for $k \neq 0$. Discarding high frequencies for $c_k$ then means removing all $a_k$ and $b_k$ coefficients whose indices are greater than some given index $n$.

## Geometric transforms

Based on fundmental properties of the discrete Fourier transform, simple rules for the change of the coefficients $c_k$ under translation, scale, and rotation immediately follow.

- **Translation**: If $u$ is a complex number then $z(t) + u$ yields the set of complex coefficients $(c_0 + u, c_1, \ldots, c_{N-1})$.
- **Scale**: If $d > 0$ then $d \cdot z(t)$ yields the set of complex coefficients $d \cdot c_k$.
- **Rotation**: Knowing that a rotation by an angle $\theta$ in the complex plane is equivalent to multiply $z(t)$ by a factor $e^{i\theta}$ leads to the set of complex coefficients $e^{i\theta} \cdot c_k$.

For scale invariance, all coefficients are divided by the absolute value of a non-zero coefficient $|c_k| = \sqrt{a_k^2 + b_k^2}$. The coefficient $c_k$ with the largest absolute value is used. Tuples $(a_k, b_k)$ are called *Harmonic Descriptors* since they apply only to a small subset of all possible non-closed curves.

**DR candidate lesion detection**

The most common signs of DR are dark lesions (micro-aneurysms, hemorrhages) and bright lesions (exudates, drusens and cotton wool spots). The presence of dark lesions and/or hard exudates (bright lesions) is indicative of early stage DR. Progression of DR also causes macular edema, neo-vessels and in later stages, retinal detachment. Many methods were proposed [4] but none of them use the radial symmetry map to detect either dark or bright lesions.

**Bright and dark candidate localization**

Micro-aneurysms are focal dilatations of retinal capillaries and appear as red dots in retinal fundus images. Wherever capillary walls are weak inside the retina, dot hemorrhage lesions are found which are slightly larger than micro-aneurysms. On rupturing it will cause intra-retinal hemorrhages. The following steps are followed in order to localize micro-aneurysms and hemorrhages on a given color fundus image: **1**) compute the symmetry image with the respective radii for the type of lesion to be detected, **2**) from the symmetry image retrieve the set of local minima, **3**) eliminate from the set of local minima the candidates located too far away from the macula, *i.e.*, candidate lesions being at a distance greater than two disk diameters from the fovea, **4**) eliminate from the set of local minima the candidates located too close from each other, **5**) use local entropy image to regroup adjacent candidates (see Fig. 2(b)), and **6**) annotate image with findings (see Fig. 1(b)).

Bright lesions or intra-retinal lipid exudates result from the breakdown of blood retinal barrier. Excluded fluid rich in lipids and proteins leave the parenchyma, leading to retinal edema and exudation. The same steps as the ones above are used to detect the bright lesions (see Fig. 1(b)).

**Deep Learning**

**Two-Step paradigm**

The two-step (*i.e.*, feature extraction and prediction) automated DR detection paradigms dominated the field of DR detection for many years. Given color fundus photography, this type of solutions often extracted visual features from the images on anatomical structures like blood vessel network, fovea and optic disc using methods put forward by the Image Processing domain such as Hough transform, Gabor filters, texture measures, *etc*. With the extracted features, an object detection or object classification algorithm like Random Forests or AdaBoost were used to identify and localize lesions. This approach forms the first phase of the DAL's detector producing feature-annotated images. The reader is refered to [15] for a review of these techniques and the performance they achieve. Table 1 reproduces some baseline results found in the literature.

**CNN**

Convolutional neural networks [6, 12] have achieved superior performance in many visual tasks, such as object classification and detection. However, the end-to-end learning strategy makes CNN representations a black box. Except for the final network output, it is difficult for people to figure out the rational behind some CNN predictions. In recent years, more and more researchers have realized that a high model interpretability is of significant values in both theory and practice and developed models with interpretable knowledge representations.

We adopted the Inception-Resnet [18] as DAL's second phase basic architecture for modelization. Inception is an architectural element developed to allow for some scale invariance in object recognition. If an object in an image is too small, like a dot hemorrhage for instance, a CNN may not

Table 1: Baseline performance for some two-step solutions

|      | Accuracy | AUC   | Sensitivity | Specificity |
|------|----------|-------|-------------|-------------|
| [9]  | -        | -     | 85.0%       | 90.0%       |
| [3]  | -        | -     | 83.9%       | 72.7%       |
| [17] | -        | 87.6% | 92.2%       | 50.0%       |

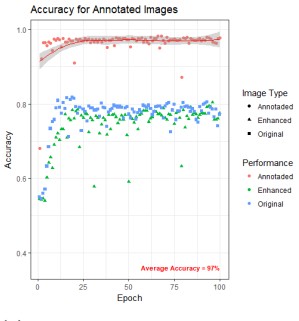

(a) Prediction accuracy of annotated image type *vs* original and enhanced types

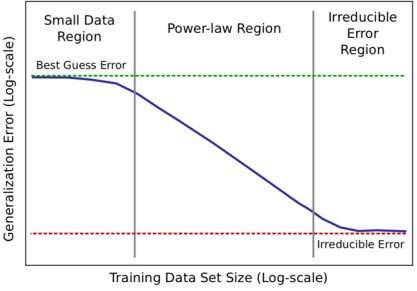

(b) Power-law scaling of generalization error

Figure 3: Prediction accuracy and generalization error power law

be able to detect it because the network filters were trained to recognize larger versions of similar objects. Inception is able to convolve through the image with various filter sizes, all at the same computational step in the network.

In order to use the pre-trained model in production, we used transfer learning and finetuning on the pre-trained Resnet-50 model [10].

**Implementation details**

A variety of cameras were used to shot the pictures, including Centervue DRS, and Topcon NW using $45^o$ fields of view. Three types of images can be produced by `DAL`'s lesion classification process: Original, Enhanced, and Annotated images. A differential experiment was conducted to check out which of these types yield the best performance when given as inputs to a CNN. The experiment was run over three equivalent datasets of 1,949 high-quality images and validated by running the model over a set of 539 images. By far, the annotated image type overruns other image types by 20 points (see Fig. 3(a)).

The original pre-processed annotated images were only used for training the network once. Afterwards, real-time data augmentation was used throughout training to improve the localisation ability of the network. During every epoch each image was randomly augmented with **1**) random rotation $-45°$ to $45°$, **2**) random horizontal, and **3**) vertical flips and random horizontal and vertical shifts.

The experiments were done on a high-end GPU, the NVIDIA K80, containing 2496 CUDA cores. It comes with the NVIDIA CUDA Deep Neural Network library (`cuDNN`) for GPU learning. Microsoft Azure Machine Learning [14] was chosen as the CNN computation back-end.

**Experimental results**

In [7], the authors presented empirical results showing how increasing training data size results in power-law scaling of generalization error and required model size to the training set for main application domains (see Fig. 3(b)). These power-law relationships, $\epsilon(m) \approx \alpha m^\beta + \gamma$ where $m$ is the train sample size and $\alpha$, $\beta$, and $gamma$ are empirical constants, hold across various model architectures, optimizers, and loss metrics. They found that for image classification the power law, as long the training set is large enough, can be expressed as $\epsilon(m) \approx 3.71 m^{-0.52}$.

From the experiments that were conducted with `Keras` over `TensorFlow` as CNN backend, the expression $\epsilon(m) \approx 9.71 m^{-0.48}$ was found to stand for the type of fundus images that we were processing. Since the accuracy *plateaus* after 5,000 images, we decided that a sample size of 7,000 would be more than enough to test the CNN with an estimated loss value of around $0.14$. The image sample was split in two datasets of 5,000 images for the training set and 2,000 for the validation set. Five experiments were run with different training/validation datasets (see Fig. 4).

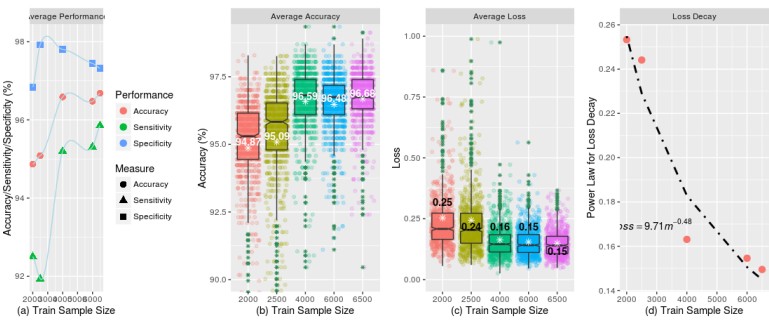

Figure 4: Power law and loss decay

### Performance

Figure 5 summarizes the performance of the algorithm in detecting referable DR (DR/No DR binary classification) on the validation data sets for fully gradable images. The algorithm achieved an average accuracy of 97.1% with an average sensitivity of 96.6% and an average specificity of 98.0% for an AUC value of 99.5%. The average loss was equal to 0.016781.

For No DR *vs.* Non-Sight Threatening DR the accuracy was 89.5% with sensitivity of 87.5% and specificity of 91.8%. No DR *vs.* Sight Threatening DR achieved an accuracy of 97.9% with sensitivity of 97.9% and specificity of 97.5%. Meanwhile, for Non-Sight Threatening DR *vs.* Sight Threatening DR the accuracy was 79.7% with sensitivity of 77.7% and specificity of 81.7%.

Furthermore, on the publicly available Messidor-2 dataset, DAL achieved a sensitivity of 92.9% and a specificity of 98.9%.

### Prior study performance comparison

Convolutional network automated DR evaluation has been previously studied by other groups. In [2], one reported a sensitivity of 96.8% at a specificity of 87% for detecting referable DR on Messidor-2. In [5], it is reported a sensitivity of 93% at a specificity of 87% on the same dataset. As for [6], a sensitivity of 87% at a specificity of 98.5% were also shown.

## Vasculature as biomarker

As diabetes affects the retina by causing deterioration of blood vessels in the retina, changes in retinal vascular caliber appear to be among the earliest changes detected in the retina of diabetic patients [8, 20]. Furthermore, previous prospective studies show that retinal signs are risk factors for ventricular enlargement, incident clinical strokes, and early and largely silent cerebrovascular

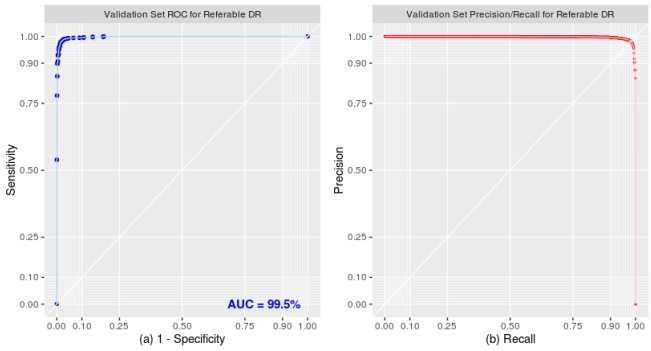

Figure 5: ROC and Precision/Recall curves

Table 2: Performance comparison

| IRIS' Dataset | Accuracy | AUC | Sensitivity | Specificity |
|---|---|---|---|---|
| DAL | **97.1%** | **99.5%** | **96.6%** | **98.0%** |

| Messidor-2 | AUC | Sensitivity | Specificity |
|---|---|---|---|
| [6] | 99.0% | 87.0% | 98.5% |
| [2] | 98.0% | 96.8% | 87.0% |
| [5] | 94.0% | 93.0% | 87.0% |
| [19] | 92.1% | 96.0% | 50.0% |
| DAL | **99.2%** | **92.9%** | **98.9%** |

changes. Thus they fulfill important requirements for being investigated as biomarkers of diabetes complications.

**Harmonic Vasculature Reconstruction**

Reconstruction of the harmonic vasculature of all 7,000 images of the dataset was done following the theory explained in the harmonic descriptors section.[2] This method removes the noise and smooth the blood vessels leaving only the important features. Experiments were run on the new derived dataset using Resnet-50 as pre-trained model. For No DR *vs.* Sight Threatening DR, an accuracy of 90.8% was achieved with an AUC of 95.8%. On the other hand, for No DR *vs.* Non Sight Threatening DR, accuracy was 64.7% with an AUC of 71.5%.

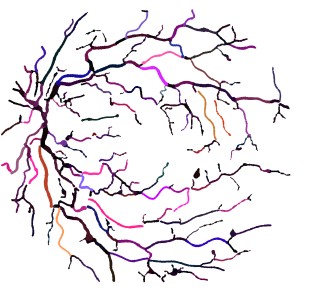

(a) Harmonic reconstruction of the vasculature in Fig. 1(a). RGB colors are respectively assigned the scale invariant values of $|(a_0, b_0)|$, $|(a_2, b_2)|$, and $|(a_1, b_1)|$

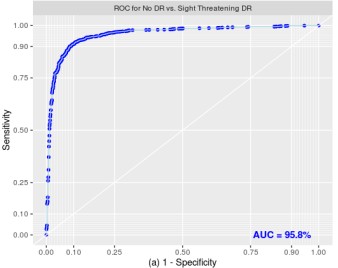

(b) ROC for No DR *vs.* Sight Threatening DR with reconstructed vasculature

Figure 6: Reconstructed vasculature and ROC curve

This shows a strong correlation between vasculature shape and later stages of DR which affects positively DAL's detection accuracy.

## Conclusion

In this evaluation of retinal fundus images from diabetic adults suffering from DR, a multi-steps paradigm solution based on Deep Learning coupled with a pattern matching pre-marking process had achieved high accuracy for detecting referable DR. Moreover, the pre-processed, pre-digested image provides the visual diagnostic report lacking from traditional CCN outcomes. Vasculature shape is a major feature of the final result.

---

[2]See Fig 6(a) where tortuosity is displayed as vessel colour and diameter as vessel width.

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
