# OpenReview forum: "Deep Annotated Learning, Harmonic Descriptors and Automated Diabetic Retinopathy Detection"
_MIDL.amsterdam/2018/Conference — Submitted to MIDL 2018_

### Review · AnonReviewer2 · 2018-05-04
**clinical relevance, weak presentation**

**Rating:** 1
**Confidence:** 2

**Review:**

This paper addresses the problem of diabetic retinopathy detection by means of a two stage procedure, involving a pre-processing step to extract features from fundus images, followed by a classification CNN to classify the images.

pros
+ clinical relevance

con
- the structure of the paper could be significantly improved
- reporting results for different classes, to better understand where errors arise, would be helpful

The presentation of the paper could be significantly improved. The sections are not numbered, making hard to follow the structure and the paragraphs are introduced in an arbitrary way.

Authors argue that feature extraction engineering results in detecting specific lesions (contrary to deep learning techniques), yet they apply this kind of handcrafted pre-processing in their pipeline.

Authors argue that pre-processing is used to make the pipeline more interpretable; could the authors clarify how the CNN becomes more interpretable given the pre-processed input?

In the algorithm descriptions, there is no connection between the parts describing the pre-processing, the Fourier descriptors theory and the CNN. It is not until page 8 that the connection becomes slightly more evident.  It might be worth summarizing how this components interact with each other when explaining each one of them separately.

The theory related to the harmonic descriptors could be significantly shortened.

Methods performing the same task and referenced in the results section should be reviewed in detail and ideally given a name, to ease the understanding of the contribution. Methods discussed in the two-step paradigm should be referenced.

It is not clear what dataset is used for results reported in Table 1. Are the results comparable to the ones reported elsewhere?

In the experiments, the difference between enhanced and annotated images is not clear.  What do you feed to your network in each case?

It is not clear whether the pipeline uses Inception-ResNet or ResNet.

What does it mean that the accuracy plateaus after 5000 images?

It seems that there is no test set, only training and validation. Is validation used to do hyper-parameter tuning? Is validation used for early stopping?

Please add references for TensorFlow and Keras.

Authors provide results for Messidor-2 dataset, which is publicly available. it would be good to have a description of this dataset. It would also be good to understand which methods you are comparing to. Are you using the same splits?

Iris dataset appears in Table 2 but is not discussed in the paper.


**Special Issue:**

No

---

### Review · AnonReviewer3 · 2018-05-08
**Disorganized and lacking important details**

**Rating:** 1
**Confidence:** 3

**Review:**

While the performance shown on the messidor dataset seemed to be state of the art there were to many problems with this paper to recommend it for acceptance.  The most significant issues is its organization and description of the lesion detection algorithm.  While the overall planned structure made sense the actual content of each section did not lead into the next and did not always contain the expected content.  This was especially true in the section describing the algorithm development.  There was a great deal of unnecessary exposition covering the basics of fourier transforms, while the actual algorithm was not discussed in sufficient detail for its implementation to be understood or replicated.  As this was the novel concept introduced by the paper this lack of detail was to important to ignore.

In addition there were multiple other inconsistencies and problems the most important being:

The authors had 250,000 exams (I'm assuming mutiple images per exams), yet only used 7000 for training and testing - why not use a great deal more images to increase the statistical significance of the validation.

The authors state that 33,428 patients (of 87,401 total) suffer from sight threatening disease yet only 8% have moderate, 3% Severe and 3% Proliferative.  They do not define sight threatening disease.

They mention in passing that the use an image quality algorithm to select which images to use for this study but provide no details concerning the algorithm or citation where such details can be found.

They claim the images are selected at random - but also say they use the quality score.  It cannot be both of these things.

They say they use an Inception-Resnet architecture, but also say they use a pretrained Resnet-50, these are not the same architectures.

There is no description of the disease prevalence and severity levels in their actual 7000 training and testing datasets.

They do not describe the actual images that are used to train the CNN.  They mention that they can produce "Original", "Enhanced", and "Annotated" but do not describe what these terms mean.

The vessel segmentation section is not sufficiently motivated in the paper and should not be included.

**Special Issue:**

No

---

### Review · AnonReviewer1 · 2018-05-09
**Lack of sufficient implementation details and novelty**

**Rating:** 1
**Confidence:** 2

**Review:**

The paper uses a two-stage process involving a – 1) clinically relevant pre-processing stage for localizing/annotating micro – aneurysms, hemorrhage and other lesions and 2) Then using a pre-trained resnet fined tuned for Diabetic Retinopathy (DR), to essentially perform a binary classification between certain DR conditions. The paper implements the above methodology on two datasets: privately held dataset and publicly available messidor-2 dataset.
The Pros and Cons of the papers are listed below:

Pros:

1. The paper presents an interesting pre-processing stage using radial symmetry transformation and entropy for identifying clinically relevant lesions in the retinal fundus images. This is the only novel addition to the paper.

2. The results on the publicly available Messidor-2 dataset are promising.

Cons:

Even though the pre-processing step presents clinical relevance, the paper lacks novelty in the area of deep learning methods with only a fine tuned resent model been used for binary DR classification. There are also certain other important issues with the current approach/paper:

1. The paper is very hard to follow – No section numbers, no flow etc.

2. Table 1 results are not sufficiently motivated and provide little or no relevance in the present context.

3. While creating a privately held dataset, the authors chose to remove the bad cases (with least quality scores) from the dataset. Without other studies on the private dataset, the strength and promise of the results cannot be fully relied upon. Also, why use only 7k images out of 250k patients. Were the rest of the exams low quality?

4. The paper presents NO experimental details on the messidor-2 dataset except the final results. Is it to be assumed that the authors used experimental set-up from one or more previous studies?

5. IRIS dataset in table 2: The authors fail of mention that this is the private dataset mentioned in the clinical dataset section.

6. The paper mentions the need/lack of interpretable methods of medical imaging for end-to-end DL methods but fails to propose how exactly the authors overcome the challenge except the medically relevant lesion identified by pre-processing stage (annotated image) being fed into the CNN instead of the raw/enhanced image. The authors need to clarify it further, if and how they tackle this issue. Also, there has been some recent work in the field of interpretable methods for DR and CV in general, which the authors are recommended to cite.

7. The later discussion of the vasculature reconstructions and biomarker discussion seems out of place and can be left out.

**Special Issue:**

No

---

### Decision · Program_Chairs · 2018-05-15
**Paper9 Acceptance Decision**

Reject